# MNCE: Multi-Hop Node Localization Algorithm for Wireless Sensor Network Based on Error Correction

**Yinghui Meng \*, Yuewen Chen, Qiuwen Zhang and Weiwei Zhang**

School of Computer and Communication Engineering, Zhengzhou University of Light Industry, Zhengzhou 450002, China; 331807040343@zzuli.edu.cn (Y.C.); 2012032@zzuli.edu.cn (Q.Z.); 2013021@zzuli.edu.cn (W.Z.)

**\*** Correspondence: yinghuimeng@zzuli.edu.cn; Tel.: +86-136-3386-3376

**Abstract:** Considering the problems of large error and high localization costs of current range-free localization algorithms, a MNCE algorithm based on error correction is proposed in this study. This algorithm decomposes the multi-hop distance between nodes into several small hops. The distance of each small hop is estimated by using the connectivity information of adjacent nodes; small hops are accumulated to obtain the initial estimated distance. Then, the error-correction rate based on the error-correction concept is proposed to correct the initial estimated distance. Finally, the location of the target node is resolved by total least square methods, according to the information on the anchor nodes and estimated distances. Simulation experiments show that the MNCE algorithm is superior to the similar types of localization algorithms.

**Keywords:** node localization; multi-hop; small hops; error correction; total least square method

---

## 1. Introduction

Wireless sensor networks (WSN) have a very wide range of practical applications [1], such as intrusion inspection, industrial automation, transportation, military, medical treatment, intelligent building, etc. WSN are commonly used for intelligent detection and response [2]. Nodes are deployed in specific environments to monitor whether the specific data of the target object is in the normal range. If an anomaly is detected, a response mechanism is triggered to respond. However, regardless of the direction to which it is applied, the information monitored by the sensor node deployed in the specified area will only work if location information is attached [3]. Otherwise, the information is meaningless. Hence, node localization is a pivotal part of any WSN.

The most direct method of localization of sensor nodes is implemented by GPS localization technologies. However, these localization methods are limited by price, volume, node lifetime and other factors of nodes [4]. Therefore, there are often difficulties in the implementing localization with accurate physical ranging technologies. We need to obtain the localization results that meet the actual demand as accurately as possible with the appropriate localization method in the case of possible low power consumption [5].

WSNs are often used in various practical requirements. We can choose the relevant measurement technology of nodes according to the actual requirements to achieve the localization function [6]. There are many classification criteria for sensor node localization algorithms, but the most common classification criterion is whether physical measurement techniques are applied to localization processes. According to this classification criterion, the node localization method can be divided into the following two types of localization algorithms [7]:

- Range-based localization algorithms require additional physical measurement techniques to complete the localization calculation. Physical measurement techniques, such as RSSI, TOA and AOA, are often used in the localization process of range-based localization algorithms [8];
- Range-free localization algorithms usually use multi-hop routing information directly obtained by sensor nodes in the network to achieve node localization [9].

The physical measurement technology used by the range-based localization algorithm requires ideal communication conditions. However, in actual application environment, we must consider factors such as node power consumption and cost [10]. If the power consumption of sensor nodes is too large, its life will be greatly shortened. As a result, it is usually not used in large-scale WSNs. The range-free localization algorithm only requires neighbor nodes to be able to communicate with each other and does not require additional distance measurement technologies, and the power consumption of WSN is greatly reduced [11]. Therefore, the range-free localization algorithm is less affected by the factors of practical application and has more extensive application [12]. However, the range-free localization algorithm has no ideal localization results, which is one of the research directions with high potential in the research of wireless sensor node localization [13]. The purpose of this study is to present a range-free localization algorithm with small positioning error and low power consumption.

## 2. Range-Free Localization Algorithms

Range-Free localization algorithms usually estimate distance between nodes by using connectivity information [14], and do not require additional physical measurement techniques to obtain node localization, so range-free localization algorithms prolong the service life of nodes [15]. However, the localization result of this type of localization algorithms has high requirements for the distribution characteristics of nodes [16]. When the network topology of WSNs is irregular, the node localization accuracy of this type of localization algorithms will be significantly reduced [17].

The well-known range-free localization algorithm includes the convex optimization algorithm, the HiRloc algorithm and the DV-hop algorithm, etc. The DV-hop algorithm obtains the minimum hop count through the message forwarding mechanism [18], and then completes the localization of the target node according to the minimum hop count and the information of anchor node. However, if we use the hop count to estimate the distance between nodes [19], there will be great localization errors, because the minimum hop count does not accurately reflect the actual distance between nodes. When there are few nodes deployed in the network or the nodes are distributed unevenly, the error of this algorithm will increase greatly.

Wang Y proposes an algorithm to calculate the single-hop correction value by using the parameters in WSNs. The LEAP algorithm [20] is proposed to calculate the single-hop correction value by using the information of anchor node locations, node communication radius, etc. However, the premise is that the sensor node satisfies a Poisson distribution. It is impossible for nodes to satisfy Poisson distribution in practical application, so LEAP algorithm is not practical.

Wu G proposes the DV-RND algorithm [21], which defines a new metric called adjustable neighborhood distance (RND). The DV-RND algorithm solves the fuzzy problem of hop distance through the proximity of nodes and the neighbor partition of nodes. However, in the WSN where nodes are randomly distributed, the localization effect of DV-RND algorithm is not ideal.

In order to obtain a higher-quality estimated distance, Shrawan K proposes the PERLA algorithm [22], which redefines the hop-size of anchor nodes, and then uses a new mathematical method to solve the equations. However, after all, the PERLA algorithm still uses hop count to estimate the distance between nodes. Therefore, in the WSN with uneven node distribution, there is still no small localization error [23].

In order to solve the problem of high localization error in current range-free localization algorithms, this study proposes the MNCE algorithm, which decomposes the multi-hop distance between nodes into several small hops. The distance of each small hops is estimated by using the connectivity information of adjacent nodes, then the error-correction rate based on the error-correction idea is

proposed to calibrate the initial estimated distance. Finally, the localization error of the MNCE algorithm is compared with the same type of localization algorithms in simulation experiments, which proves the superiority of the proposed algorithm.

## 3. MNCE Algorithm

Regardless of the type of sensor node localization algorithm, the localization process can be roughly decomposed into the following three stages [24]:

1.  Distance estimation: We can get the estimated distance between nodes by the information that can be directly obtained by sensor nodes such as arrival time, hop number information and connected information;
2.  Initial localization: According to the estimated distance of the previous stage, the corresponding estimation algorithm is selected to complete the initial localization of the target node;
3.  Calibration localization: According to the information of the first two stages, the redundant information is eliminated, and the corresponding optimization algorithm is selected to optimize and calibrate the estimated location of target nodes.

The MNCE algorithm divides the multi-hop distance into several small hops; then calculates the small hops one by one to get the estimated distance. The algorithm is described from these three stages.

### 3.1. Distance Estimation

We suppose that the quantity of nodes in a WSN is $N$, the quantity of anchor nodes is $n$. The communication radius of all nodes is $R$, nodes get their own information with neighbor nodes by forwarding messages. The neighbor relationship model is as follows:

$$Wi = \left\{ j \middle| j \neq i \&\& d_{ij} \leq R \right\} \tag{1}$$

The $i$ and $j$ in Formula (2) symbolize two sensor nodes; $d_{ij}$ represents the Euclidean distance between the two nodes. The general idea of this type of localization algorithms is to propose a measure which is positively correlated with the distance between nodes according to the node distribution and node connectivity information, and then to represent the measure as accurately as possible according to the locality of the node distribution. As an example of DV-hop algorithm: The estimated distance is calculated by the minimum hop count and the single hop correction.

As shown in Figure 1, this localization method will have a large error. The hop count between nodes will be defined as one hop, if the actual distance between the two sensor nodes is between 0 and $R$. We propose the MNCE algorithm, which uses the relationship of the distance between adjacent nodes and the area of communication overlapped region between adjacent nodes to obtain the estimated distance between nodes more accurately.

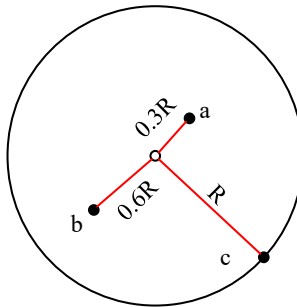

**Figure 1.** Hop count between the nodes connected by the three red lines is one hop, but the actual distance between them is quite different.

We divide the estimation distance process into several steps: first, we divide the multi-hop distance into the accumulation of multiple single-hop distances; second, we estimate the distance for each single-hop distance. The node $i$ and node $j$ in Figure 2 are neighbor nodes with one hop in the sensor network. The black solid points in Figure 2 represent the different neighbor nodes of the two nodes, and the red solid points represent the common neighbor node of two nodes.

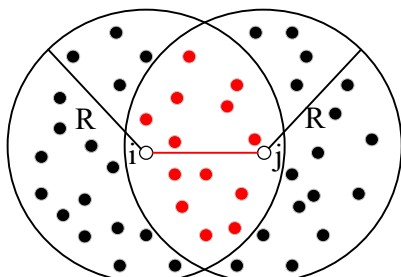

**Figure 2.** Communication between two nodes with one hop.

The area of the communication overlap region $A_{ij}$ is obtained by geometric calculation:

$$A_{ij} = 2R^2 \cos^{-1}(\frac{d_{ij}}{2R}) - d_{ij}\sqrt{R^2 - \frac{d_{ij}^2}{4}} \tag{2}$$

The $d_{ij}$ in Formula (2) is the distance between the two nodes, so we can get the area ratio between node communication overlap region and node communication region as shown in Formula (3):

$$\frac{A_{ij}}{\pi R^2} = \varphi(d_{ij}) = \frac{2}{\pi}\cos^{-1}(\frac{d_{ij}}{2R}) - \frac{d_{ij}}{\pi R}\sqrt{1 - (\frac{d_{ij}}{2R})^2} \tag{3}$$

$$d_{ij} = \varpi(\frac{A_{ij}}{\pi R^2}) \tag{4}$$

There are two unknown terms in Formula (3): Area ratio and $d_{ij}$, so we cannot work out the $d_{ij}$ according to one formula. Formula (4) is the inverse function of Formula (3). We assume that there are a large number of sensor nodes in the WSN, and the nodes deployment characteristics in the local region of the node communication range are approximately the same. In addition, the ratio of the two regions is approximately equal to the ratio of the quantity of nodes in two regions. The MNCE algorithm does not require additional measurement techniques, as long as the neighbor nodes can communicate. Therefore, this type of localization algorithms has low power consumption and nodes can be deployed with the high density in such algorithms. The node distribution of local neighboring regions is approximately the same in WSNs with the node high density deployment. Therefore, we can approximate the area ratio of the two regions by the ratio of the quantity of nodes in the two local neighboring regions. Therefore, the distance $d_{ij}$ between neighbor nodes can be calculated using Formula (3):

$$\frac{A_{ij}}{\pi R^2} = S = \varphi(d_{ij}) \approx \frac{|W_i \cap W_j|}{\max(|W_i|+1, |W_j|+1)} \tag{5}$$

$$d_{ij} = \varpi(S) \approx \varpi(\frac{|W_i \cap W_j|}{\max(|W_i|+1, |W_j|+1)}) \tag{6}$$

The more nodes are deployed in the network, the closer the proportion of the quantity of nodes is to the area ratio of the regions where the node is located. Therefore, in Formula (6), we choose the maximum value of the quantity of nodes in the communication area of two nodes to replace the denominator of the area ratio.

Suppose the communication path between two nodes is shown in Figure 3. The estimated distance is the superposition of each hop distance. We assume the minimum hop count is $n$ and the distance between nodes of the i-th hop is $d_i$, obviously the initial estimated distance is calculated using the above method:

$$\widetilde{d_{ij}} = d_1 + d_2 + \cdots + d_n \tag{7}$$

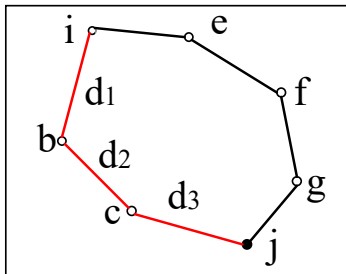

**Figure 3.** The two broken lines between node *i* and node *j* represent two communication paths, the red line represents the shortest path, and d1, d2 and d3 are the distance of each hop.

However, the shortest path of communication is not a straight line under most circumstances. When the shortest path of two nodes is a tortuous route, the error will gradually become larger when the estimated distance is calculated by using a single hop distance accumulation method. As shown in Figure 4 below, the actual distance $d_{ij}$ between the two hops is much less than $d_{ik} + d_{kj}$.

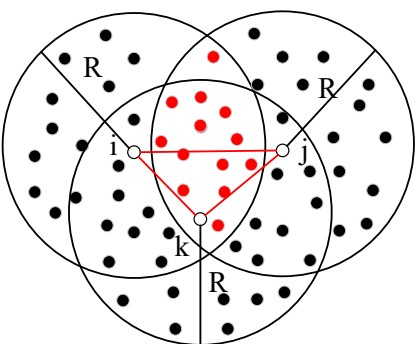

**Figure 4.** Communication between two nodes with two hops.

In order to find out the estimated distance between nodes more accurately and reduce the error, we should split the multi-hop distance between nodes up small hops with as little as possible. The relationship between distance $d_{ij}$ and area $A_{ij}$ must satisfy the Formula (2) if the distance between nodes is calculated by the method in this study. When the hop count is greater than 2, there is no overlapped area in the communication area between two nodes. Therefore, it is reasonable to divide the multi-hop distance into multiple accumulations in units of two hops. The node distance and area of two hops also apply to the function $\varphi(d_{ij})$:

$$\frac{A_{ij}}{\pi R^2} = S = \varphi(d_{ij}) \approx \frac{\left|W_i \cap W_j\right|}{\max(\left|W_i\right|+1, \left|W_k\right|+1, \left|W_j\right|+1)} \tag{8}$$

We assume the minimum hop count is $n$. When $n$ is even, the estimated distance is shown in Formula (9). when $n$ is odd, the estimated distance is shown in Formula (10):

$$\widetilde{d_{ij}} = \sum_{i=1}^{n/2} \omega\left(\frac{\left|W_i \cap W_{i+2}\right|}{\max(\left|W_i\right|+1, \left|W_{i+1}\right|+1, \left|W_{i+2}\right|+1)}\right) \tag{9}$$

$$\widetilde{d}_{ij} = \sum_{i=1}^{(n-1)/2} \omega\left(\frac{\left|W_i \cap W_{i+2}\right|}{\max(\left|W_i\right|+1, \left|W_{i+1}\right|+1, \left|W_{i+2}\right|+1)}\right) + d_{last} \tag{10}$$

### 3.2. Error-Correction Rate for the Estimated Distance

We can find the initial estimated distance between neighboring nodes through the probability distribution and function calculation. However, when the hop count is many, the initial estimated distance will lack accuracy and there will be a certain error. If we can calculate the estimated distance error and then correct the initial estimated distance properly with this, the estimated distance error will be greatly reduced. The node distribution of WSNs is random and the neighboring nodes have the approximate deployment environment. Therefore, we should study the locality of node distribution of WSNs. The actual location of anchor nodes can be measured by the device or known in advance. Therefore, the actual distance between anchor nodes can be obtained. We use the above distance estimation method to calculate the estimated distance between anchor nodes. Then the difference between the two distances is the distance estimation error, and the estimation error provides a reference for distance estimation for adjacent unknown nodes. Therefore, the error-correction rate of initial distance estimation is proposed:

$$\alpha_i = \frac{\sum\limits_{j \in M, j \neq i} d_{ij}}{\sum\limits_{j \in M, j \neq i} \widetilde{d}_{ij}} \tag{11}$$

$$d_{ij} = \sqrt{(x_i - x_j)^2 + (y_i - y_j)^2} \tag{12}$$

Formula (11) is the error-correction rate of the anchor node $i$. $M$ is the set of anchor nodes, the actual coordinates of anchor node $i$ are $(x_i, y_i)$ and $d_{ij}$ is the actual distance between the two anchor nodes.

Then the final estimated distance between two nodes is as follows (In Formula (13), $i$ represents the anchor node and $j$ represents the unknown node):

$$\overline{d}_{ij} = \alpha_i \times \widetilde{d}_{ij} \tag{13}$$

### 3.3. Node Location Method Based on Total Least Squares

It has proved to be an effective method to locate the target node by using the location of anchor nodes and the estimated distance in WSNs. We suppose that the coordinates of the target node is $(x, y)$, and the estimated distances to all anchor nodes are $\overline{d}_1, \overline{d}_2, \cdots \overline{d}_n$ ($n \geq 3$). As shown in Figure 5 below, if the estimated distances are accurate; take more than three anchor nodes as the center of the circle and the intersection of the circles with the estimated distance as the radius is the location of the target node.

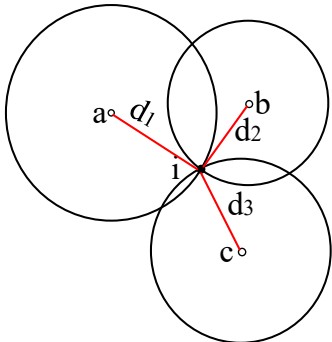

**Figure 5.** Target node localization.

According to the distance relationship between nodes, we can obtain a nonlinear equation. By subtracting the *n-th* expression from the first *n-1* expressions in the nonlinear equations, the equations can be simplified into linear overdetermined equations:

$$Ax = b \tag{14}$$

$$A = \begin{bmatrix} 2(x_1 - x_n) & 2(y_1 - y_n) \\ 2(x_2 - x_n) & 2(y_2 - y_n) \\ 2(x_3 - x_n) & 2(y_3 - y_n) \\ \vdots & \vdots \\ 2(x_{n-1} - x_n) & 2(y_{n-1} - y_n) \end{bmatrix} \tag{15}$$

$$x = \begin{bmatrix} y \\ x \end{bmatrix} \tag{16}$$

$$b = \begin{bmatrix} x_1{}^2 - x_n{}^2 + y_1{}^2 - y_n{}^2 + \bar{d}_n{}^2 - \bar{d}_1{}^2 \\ x_2{}^2 - x_n{}^2 + y_2{}^2 - y_n{}^2 + \bar{d}_n{}^2 - \bar{d}_2{}^2 \\ x_3{}^2 - x_n{}^2 + y_3{}^2 - y_n{}^2 + \bar{d}_n{}^2 - \bar{d}_3{}^2 \\ x_{n-1}{}^2 - x_n{}^2 + y_{n-1}{}^2 - y_n{}^2 + \bar{d}_n{}^2 - \bar{d}_{n-1}{}^2 \end{bmatrix} \tag{17}$$

If the known data in this equation group are accurate, the node location calculation can be completed by the least square method. However, in practical application, the estimated distance is not accurate, even the anchor node in the network has errors due to localization technology or deployment problems. Therefore, there is an error in both the coefficient matrix $A$ and the observation vector $b$ in Formula (14). If the error matrix of the coefficient matrix $A$ is $E_A$ and the error vector of the observation vector $b$ is $E_b$, the actual formula is shown in Formula (18):

$$(A + E_A)x = b + E_b \tag{18}$$

Considering this problem, this study chooses the total least square method to complete the localization calculation. The total least square method is an advanced least square method [25], which comprehensively considers the error of the coefficient matrix $A$ and the observation vector $b$ and has high calculation accuracy and feasibility. Compared with the least square method, the method does not need a lot of sample data. The main idea of the method is to minimize the coefficient matrix and the observation vector. That is, the $F$ norm of the disturbance matrix is the smallest, which can be solved by singular value decomposition (SVD). Singular value decomposition SVD can extract the important structure information hidden in the matrix. More important, it can also reduce the dimension of the matrix. We construct an augmented matrix and perform singular value decomposition:

$$C = \begin{bmatrix} A \vdots b \end{bmatrix} = U \begin{bmatrix} \Sigma & 0 \\ 0 & 0 \end{bmatrix} v^H \tag{19}$$

In Formula (19) $U = (u_1, u_2, \cdots, u_{n-1})$, $\Sigma = \text{diag}(\sigma_1, \sigma_2, \cdots, \sigma_r)$, $V = (v_1, v_2, \cdots, v_{k+1})$, $k$ is the number of dimensions to be solved, in this study $k$ is 2. Assume that the smallest non-zero singular value $\sigma_r$ corresponds to the vector in the right singular matrix $V$ as follows:

$$v_r = (v_{1,r}, v_{2,r}, \cdots, v_{k+1,r})^H \tag{20}$$

The final result is:

$$\hat{x} = -\frac{1}{v_{k+1,r}} (v_{1,r}, v_{2,r}, \cdots, v_{k,r})^H \tag{21}$$

## 4. Simulation Experiment

We use MATLAB to simulate experiments to verify the superiority of the MNCE algorithm. We randomly deploy *N* nodes in an area 100 meters long and 100 meters wide. These nodes form WSNs through self-organization without any additional physical measurement techniques. In Figure 6 below, the black circular points represent unknown nodes, and the blue star points represent anchor nodes.

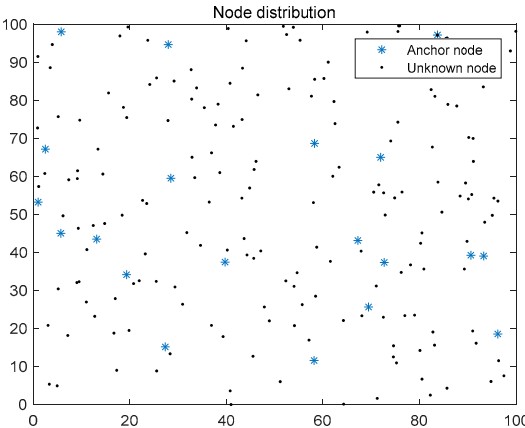

**Figure 6.** Node distribution.

In order to make the simulation experiment more objective, we conduct each experiment 50 times under relatively independent conditions and the final experimental results are the average of 50 experiments. In this section, we compare the MNCE algorithm with the three algorithms of DV-hop, DV-RND and PERLA under the same circumstances.

### 4.1. Performance Indicators

We usually judge an algorithm by error. In the process of node localization, the quality of the distance estimation directly affects the final localization result. Therefore, in order to make the experimental results more representative, two performance indicators are defined in this study: estimation distance error and localization error:

$$ADE = \frac{1}{\sum\limits_{i=1}^{N} |w_i|} \sum_{i=1}^{N} \sum_{j \in Wi} \left| d_{ij} - \bar{d}_{ij} \right| \tag{22}$$

$$d_{ij} = \sqrt{(x_i - x_j)^2 + (y_i - y_i)^2} \tag{23}$$

$$APE = \frac{1}{(N-n)} \sum_{i=1}^{N-n} \sqrt{(x_i - \bar{x}_i)^2 + (y_i - \bar{y}_j)^2} \tag{24}$$

The above formulas are the absolute estimated distance error *ADE* and the absolute localization error *APE*, but the algorithm error has a great relationship with the communication radius. When the communication radius is uncertain, it is meaningless to compare the localization error alone. Therefore, introducing the relative error of the radius can describe the MNCE algorithm performance more accurately. As shown in the following formula, they are the estimated distance error *RADE* and the localization error *RAPE*, respectively:

$$RADE = \frac{1}{R \sum\limits_{i=1}^{N} |w_i|} \sum_{i=1}^{N} \sum_{j \in W_i} \left| d_{ij} - \bar{d}_{ij} \right| \tag{25}$$

$$RAPE = \frac{1}{R(N-n)} \sum_{i=1}^{N-n} \sqrt{(x_i - \overline{x}_i)^2 + (x_i - -\overline{y}_j)^2} \tag{26}$$

There are many factors will affect the localization results of target nodes. In this study, we select three important factors. Next, we will compare the pros and cons of the MNCE algorithm through simulation experiments for these three factors. For the convenience of recording, in the following content, *N* is the total number of nodes, *POA* is the proportion of anchor nodes and *R* is the communication radius of the nodes.

### 4.2. Impact of the Error Correction Rate on RADE

In order to get more accurate estimation results of distance between nodes, the error-correction rate is proposed in this study. In this section, the effectiveness of the error-correction rate is tested by simulation experiments. First, we number the unknown nodes: *1–270* and then take the average of the *RADE* of each three nodes as an experimental result. The final experimental results are shown in Figure 7. The average estimated distance errors of the MNCE algorithm and the MNCE (NC) algorithm are 11.3073 and 12.8912, respectively. The experiment in this section proves the advanced nature of the error-correction rate (MNCE (NC) is the algorithm proposed in this study without the error-correction rate correction).

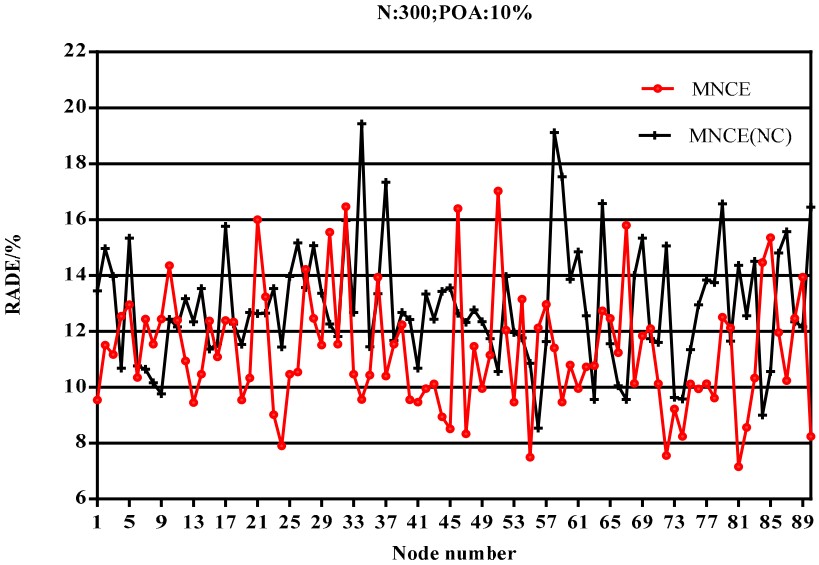

**Figure 7.** Impact of the error-correction rate on estimated distance error (*RADE*).

### 4.3. Impact of Node Communication Radius on RADE

As we all know, the accurate estimation distance is the premise of a high-quality node localization algorithm. In the distance estimation stage, the quantity of nodes and the communication radius are two factors that greatly affect the accuracy of the estimated distance. The four algorithms compared in this study are all range-free localization algorithms with low power consumption. Therefore, this section only conducts experiments on the node communication radius to observe its influence on the estimated distance error (*RADE*).

The experimental results are shown in Figure 8, when *R* is less than 20 m, the estimated distance error of the four localization algorithms compared in this study decreases with the increase of *R*. However, when *R* is larger than 20 m, the estimation distance error of the DV-hop algorithm increases with the increase of *R*. DV-hop uses the hop count to estimate the distance, if *R* increases too much, it is beyond the appropriate range, the single-hop correction value will often have side effects. However,

in the range of communication radius change, the calculation results of the MNCE algorithm are always superior to the other three.

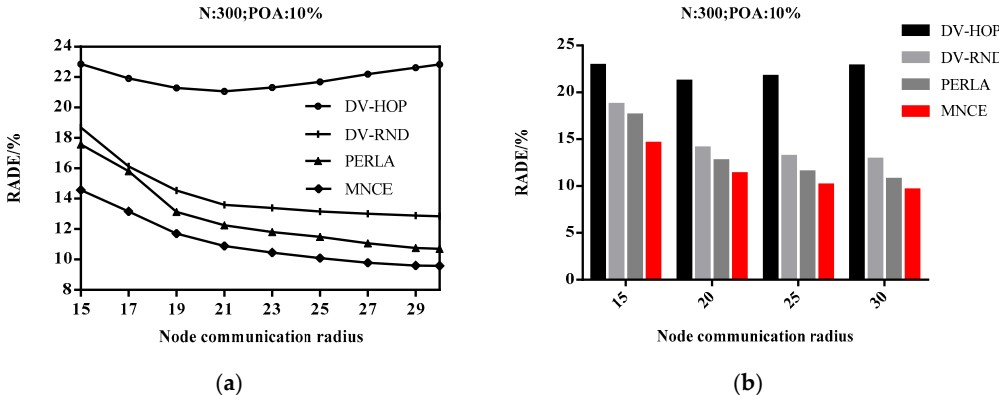

**Figure 8.** Impact of communication radius on RADE. (**a**) broken line diagram of the experimental results; (**b**) histogram of experimental significance values.

### 4.4. Impact of the Total Number of Nodes on RAPE

The experimental results are shown in Figure 9. When *N* increases from 200 to 60 z0, the *RAPE* of the four algorithms in this study shows a significant decrease. However, no matter how much *N* is, the *RAPE* of the MNCE algorithm in this study is smaller than the other three localization algorithms. The more nodes are deployed in WSNs, the more uniform node distribution appears in the local area, and the closer the shortest path is similar to the real distance path. Therefore, the proposed MNCE algorithm has better localization results when it is applied to large-scale WSNs.

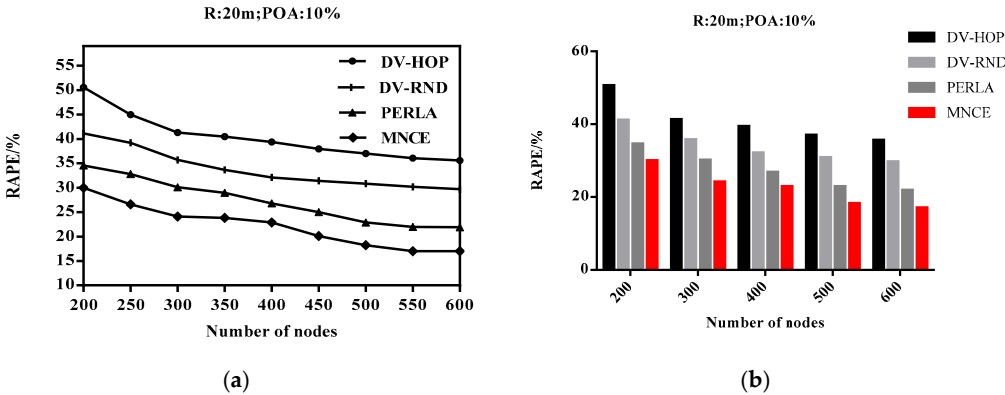

**Figure 9.** Impact of total number of nodes on *localization error (RAPE)*. (**a**) broken line diagram of the experimental results; (**b**) histogram of experimental significance values.

### 4.5. Impact of Node Communication Radius on RAPE

As shown in Figure 10, the *RAPE* of four localization algorithms decreases with the increase of *R*. The larger *R*, the closer the ratio of the number of nodes in two communication areas is to the ratio of the area of two communication areas, so the smaller the error of the MNCE algorithm is. As the experimental results show, the *RAPE* of the MNCE algorithm at different node communication radius is smaller than other algorithms, which proves the effectiveness of the MNCE algorithm.

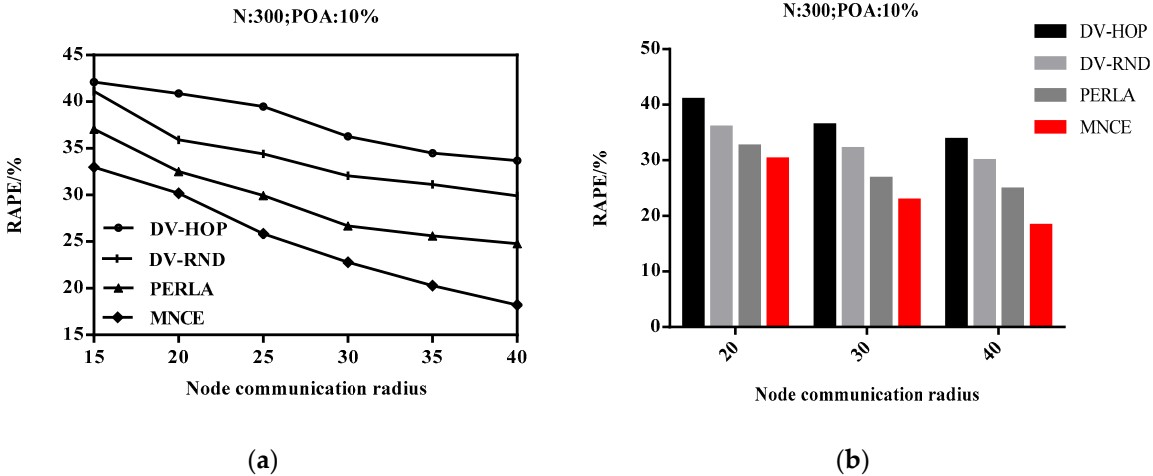

**Figure 10.** Impact of communication radius on *RAPE.* (**a**) broken line diagram of the experimental results; (**b**) histogram of experimental significance values.

### 4.6. Impact of the Proportion of Anchor Nodes on RAPE

With the increase of *POA*, the *RAPE* of the four localization algorithms are reduced. The increase of anchor nodes has a relatively small influence on the distance estimation stage, but it has a great influence on the localization estimation stage which provides the possibility for localization calibration. Therefore, the localization error can be greatly reduced. As shown in the experimental results of Figure 11, the *RAPE* of the MNCE algorithm is superior to the other three at different *POA*, which proves the effectiveness of the MNCE algorithm.

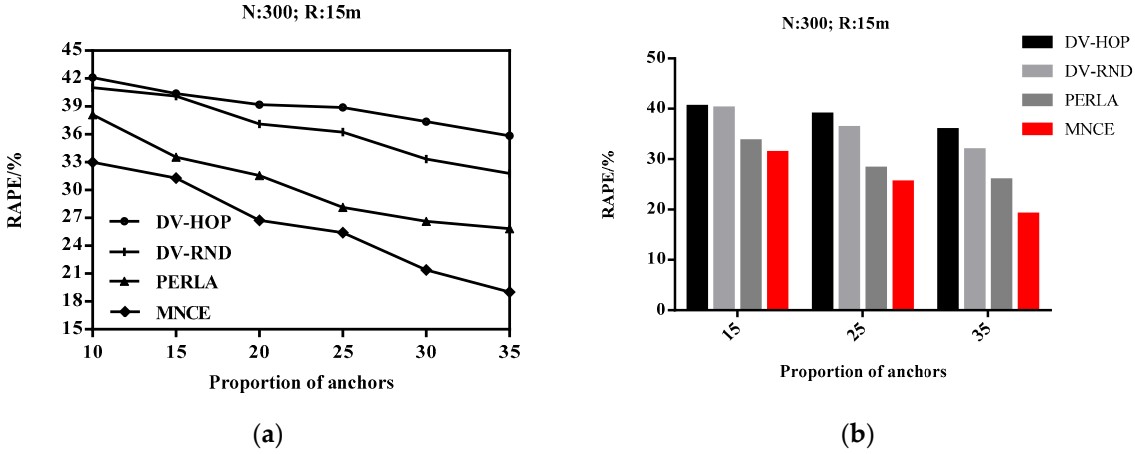

**Figure 11.** Impact of *POA* on *RAPE.* (**a**) broken line diagram of the experimental results; (**b**) histogram of experimental significance values.

## 5. Conclusions

This study proposes the MNCE algorithm based on the error correction. This algorithm decomposes the multi-hop distance between nodes into several small hops. The distance of each small hop is estimated by using the connectivity information of adjacent nodes, then the error-correction rate based on the error-correction idea is proposed to calibrate the estimated distance. Next, the nonlinear equations are set up by the location of anchor node and the estimated distance, and the total least square method is solved by singular value decomposition method. The simulation experiments show the superiority of the MNCE algorithm.

**Author Contributions:** Data curation, Y.M. and Y.C.; formal analysis, Q.Z.; funding acquisition, Y.M.; investigation, Y.C.; methodology, Y.C. and Q.Z.; project administration, Y.M. and W.Z.; supervision, Y.M. and W.Z.; writing—original draft, Y.C.; writing—review & editing, Y.M. and Y.C. All authors have read and agreed to the published version of the manuscript.

**Funding:** This research was funded by National Natural Science Foundation of China, grant number 61501405, Science and Technology Planning Program of Henan Province, grant number 202102210398.

**Acknowledgments:** This work is supported by the National Natural Science Foundation of China (No. 61501405), the Science and Technology Planning Program of Henan Province (No. 202102210398).

**Conflicts of Interest:** The authors declare no conflict of interest.

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
