# Peer review of "MNCE: Multi-Hop Node Localization Algorithm for Wireless Sensor Network Based on Error Correction"

_information, doi:10.3390/info11050269_

Round 1

Reviewer 1 Report

Authors present an approach to reduce large error in range free localization. Multi-hop distance is decomposed into small hops for this purpose and connectivity information is used to estimate the distance.   The manuscript is written well and easy to follow.  

I have the following comments for the manuscript:

  1. As shown in Figure 7 that if we increase node communication radius from 15 to 17 the error (RADE %) is reduced. What will happen if we increase it further? Is there any optimal number for communication radius?
  2. Figure 7 and Figure 8 look contradictory. If we increase communication radius distance error will be reduced. If we increase number of nodes, localization error will be reduced. But if we increase nodes, it will decrease the communication radius between nodes. If so, it will increase distance error. Please explain.
  3. English language needs improvement.

Reviewer 2 Report

Comment 1.

The method seems to be simple for implementation and requires little computational and measurement effort – which is important for low power wireless sensor network (WSN). Presented results have been compared to a few other methods and are promising.

However, Authors assumed uniform nodes distribution (and constant among network) but, at the same type, have criticized other methods for the same assumption:

  • row 138: "We assume that there are a large number of sensor nodes in the WSN, the nodes distribution in the local region of the node communication range can be approximately uniformly distributed."
  • row 178: "The node distribution of WSNs is random, and the neighboring nodes have the same distribution."
  • row 60: "When the local distribution of nodes is uniform, the accuracy of this localization results will be very low [16]. When the network topology of WSNs is irregular, the node localization accuracy of this type of localization algorithms will be significantly reduced [17].".

It would be interesting to verify how the presented method behaves for case of irregular WSN. Even, if the results were worse, it would be good idea to test proposed solution for typical, real-life cases. Maybe, the Authors' method behaves better for irregular WSN than the other methods for the same case?

Comment 2.

The presented ideas are interesting, but lack detailed description:

  • how multiple hop positioning is transformed into single hop positioning? There is no comparison between these cases.
  • Authors mention about “Error correction rate for the estimated distance” (subchapter 3.2), but the correction itself (error before and after correction) is not compared (presented).

Comment 3.

The subchapter 3.3 presents “Node location method based on total least squares” where standard trilateration (or multilateration) positioning is used:

  • row 194: "As shown in Figure 5 below, if the estimated distance is accurate; take more than three anchor nodes as the center of the circle, and the intersection of the circles with the estimated distance as the radius is the location of the target node."

And “singular value decomposition (SVD)” is used to minimize positioning uncertainty (if distance measurements are uncertain). This is fine, of course, since it works. However, other techniques would be worth noticing, perhaps with less computational demands:

  • Chruszczyk L., Statistical Analysis of Indoor RSSI Read-outs for 433 MHz, 868 MHz, 2.4 GHz and 5 GHz ISM Bands, International Journal of Electronics and Telecommunications (IJET), 2017, Vol. 63, No. 1, pp. 33-38, DOI 10.1515/eletel-2017-0005, e-ISSN 2300-1933.
  • Grzechca D., Paszek K., Short-term positioning accuracy based on mems sensors for smart city solutions, Metrology and Measurement Systems, t. 26, nr No 1. Polish Academy of Sciences Committee on Metrology and Scientific Instrumentation, s. 95–107, 2019, doi: 10.24425/mms.2019.126325.

Comment 4.

Positioning accuracy depends on anchor nodes number and distribution. It would also be worth noticing alternative low-power techniques for anchor nodes (range based localization):

  • Chruszczyk L., Zajac A., Comparison of indoor/outdoor, RSSI-based positioning using 433, 868 or 2400 MHz ISM bands, International Journal of Electronics and Telecommunications (IJET), 2016, Vol. 62, No. 4, pp. 395-399, DOI 10.1515/eletel-2016-0054, ISSN 2081-8491, e-ISSN 2300-1933.
  • Grzechca D., Hanzel K., The Positioning Accuracy Based on the UWB Technology for an Object on Circular Trajectory, International Journal of Electronics and Telecommunications, vol. 64, No 4., Polish Academy of Sciences Committee of Electronics and Telecommunications, p. 487–494, 2018, doi: 10.24425/123550.

Comment 5.

Some confusing expressions have been found:

  • row 59: "However, the localization result of this type of localization algorithms is decided by the distribution characteristics of nodes. When the local distribution of nodes is uniform, the accuracy of this localization results will be very low [16]. When the network topology of WSNs is irregular, the node localization accuracy of this type of localization algorithms will be significantly reduced [17]." - uniform or irregular nodes distribution gives better location accuracy (smaller error)?
  • row 226: "In the Figure 6 below, the black circular points represent anchor nodes, and the blue star points represent unknown nodes." - I think there is a mistake in description of node types.

Comment 6.

Please improve your language and correct typing:

  • some sentences are unnecessarily repeated (copied), e.g. "has a very wide range of practical applications" - twice
  • row 56: remove extra initial dot in chapter title: ".2. Range-Free Localization algorithm"
  • row 86: the language needs improvements, e.g. "Considering some problems existing in the current node localization algorithms, we propose the MNCE algorithm in this paper. Which decomposes the multi-hop distance between nodes into several small hops."
  • row 144: language: "The local node distribution is approximately uniform in WSNs with node high density deployment. Therefore. We can approximate the ratio of the areas of the two regions by the ratio of the quantity of nodes in the two regions."
  • row 146, language: "Furthermore. the distance between neighbor nodes can be obtained."

Reviewer 3 Report

This study is to explore multi-hop node localization algorithm for
wireless sensor network in terms of error correction. Topic is timely and overall study processes are valid and proper. There are some comments to improve te quality of the paper.

  • Add "The purpose of this study is to present ...." in second part of the introduction section.
  • Add some data validity in simulation settings. 
  • Add some statistical significance value among different methods in figures 7, 8, 9 and 10 
